# The Influence of Chestnut Flour on the Quality of Gluten-Free Bread

**Katarzyna Marciniak-Lukasiak [1,*]**, **Patrycja Lesniewska [1]**, **Dorota Zielińska [2]**, **Michal Sowinski [1]**, **Katarzyna Zbikowska [3]**, **Piotr Lukasiak [4]** and **Anna Zbikowska [1]**

1.  Institute of Food Sciences, Faculty of Food Assessment and Technology, Warsaw University of Life Sciences (WULS-SGGW), Nowoursynowska St. 159c, 02-776 Warsaw, Poland
2.  Department of Food Gastronomy and Food Hygiene, Institute of Human Nutrition Sciences, Warsaw University of Life Sciences (WULS-SGGW), Nowoursynowska 159c, 02-776 Warsaw, Poland
3.  Faculty of Medicine, Medical University of Warsaw, Zwirki i Wigury St. 61, 02-091 Warsaw, Poland
4.  Institute of Computing Science, Faculty of Computing and Telecommunications, Poznan University of Technology, Piotrowo 2, 60-965 Poznan, Poland
*   Correspondence: katarzyna_marciniak_lukasiak@sggw.edu.pl; Tel.: +48-22-59-37548

**Abstract:** Gluten-free bread is the basis of an elimination diet in the case of many glucose-related diseases. The quality of this bread differs significantly from traditional products; therefore, it is necessary to conduct research aimed at improving the quality of this type of product. The aim of the study was to determine the effect of the addition of chestnut flour and the method of packaging on the quality of gluten-free bread. The addition of chestnut flour (partially replacing corn starch) was used in the amount of 5, 10, 15 and 20% of the total weight of the concentrate. The influence of the storage method on the quality of the tested bread was examined after 7, 14 and 21 days from baking. The refrigerated breads were packed using PA/PE barrier foil with air and vacuum (58%) and were stored in room temperature ($22 \pm 2$ °C). Water content, texture and color were determined, and sensory evaluation and microbiological analysis were performed. As a result of the conducted research, we observed that the addition of chestnut flour to the recipe affects significantly ($p < 0.05$) the texture of the finished product, reducing the hardness and increasing the elasticity and cohesiveness of the bread crumb. The use of chestnut flour in an amount of up to 10% increases significantly ($p < 0.05$) the volume of the resulting loaves. Microbiological research has indicated vacuum packaging as a better way to protect and store gluten-free bread. For practical use in future production, it is recommended to replace corn starch in gluten-free breads by no more than 10% by chestnut flour.

**Keywords:** celiac disease; gluten-free bread; chestnut flour; texture; sensory analysis; vacuum packaging

## 1. Introduction

Celiac disease (CD), defined as chronic autoimmune gluten intolerance, is becoming an increasingly important health problem in modern medicine in developed countries [1,2]. The occurrence of CD is strongly related to genetic factors, among which HLA class II plays a major role—HLA-DQ2 heterodimers are expressed in over 90% of patients, where the remnant express HLA-DQ8 [3,4]. So far, the only effective treatment option is to follow a gluten-free diet throughout the entire life [5,6].

Gluten-free products are the basis of the elimination diet of patients suffering from gluten-dependent diseases, which include, among others: celiac disease, nonceliac hypersensitivity to gluten, Duhring's disease and wheat allergy [7,8]. According to the research, celiac disease affects about 1% of the population, and the incidence of this disease is gradually increasing. Failure to follow a strict diet ultimately leads to the disappearance of intestinal villi, which in turn results in the malabsorption of nutrients from food. The

limited absorption of ingredients necessary for the proper functioning of the body may cause various clinical symptoms [2].

Bread is one of the basic ingredients in the daily human diet. It is the main source of carbohydrates and provides many valuable nutrients, including B vitamins and fiber to support the proper functioning of the intestines. Due to the changing eating habits and the increase in various types of food allergies and intolerances, newer recipes and technological solutions are sought to expand the range of products available on the market [9–11].

According to the Codex Standard [12], gluten is the protein fraction of wheat, rye, barley, oats or their hybrids and derivatives that some people cannot tolerate. The water-insoluble prolamins and glutelins (collectively referred to as gluten) typically make up 70–80% of the cereal grain protein. They are the most important cereal proteins from a technological point of view. In this sense, gluten is a specific structure, a viscoelastic gel, which is responsible for creating the correct structure of the finished product with unique properties. As a result of absorbing large amounts of water, gluten swells, giving the dough its appropriate characteristics, such as flexibility, plasticity, stickiness, cohesiveness and elasticity. It allows dough to obtain a thin and spongy structure with fine pores, which is fixed during baking [10].

Therefore, the production of bread from gluten-free raw materials is a major technological challenge. The lack of gluten causes many problems in the preparation of the dough, significantly affecting its rheological properties and the quality of the final product. The obtained products are often characterized by a poorer structure, color and porosity of the crumb, smaller volume, stickiness and considerable brittleness in comparison to the regular one. They are also characterized by a short shelf life and worse sensory features compared to traditional baking [13].

In order to improve the rheological properties of the dough, which will allow for the proper forming of the billets and obtaining the appropriate properties of the finished product, many studies are carried out to improve the recipes of gluten-free bread. In order to replace gluten, which is a structure-forming factor, various substances are used to support the proper development of the dough, emulsifiers, as well as texturing and thickening substances. The most commonly used ingredients are hydrocolloids [14,15], which include, inter alia, guar gum, xanthan gum, Locust bean gum, pectin, carboxymethyl cellulose and hydroxyethyl cellulose. These substances also have the ability to bind water, gel and act as stabilizers. Enzymes (transglutaminases, amylases and proteases) and exopolysaccharides, which are produced by lactic acid bacteria, are also increasingly used, which allows for the elimination of chemical substances forming the structure [16–20].

Due to their specific recipe composition, gluten-free products have a lower nutritional value and a higher glycemic index compared to their traditional counterparts. In order to improve the properties of gluten-free bread, enrichment with the addition of raw materials of plant and animal origin is used. This will increase its nutritional value, obtain a high-quality product and gain consumer recognition [21,22]. An example of such an additive is chestnut flour, which also positively influences the physical and chemical properties of food products like cookies [23], pasta [24] or bread [23,25–28]. Chestnuts are a good source of antioxidants and minerals such as potassium and magnesium, which are associated with reducing the risk of cardiovascular disease and stroke [26]. Chestnut flour, on the other hand, is a good functional ingredient and may increase the content of some nutrients, positively affecting the physical and nutritional properties of cereal products [23] and the quality features of the finished product [27,28]. In practice, however, the amount of the additive used often has to be limited, because too much chestnut flour may reduce the quality of the finished product (the addition of chestnut flour creates a darker and often harder product, but, especially for gluten-free bread, improved dough workability, texture, color and flavor) [9,19,23,29].

Demand for gluten-free products continues to increase, with a global market of USD 21.61 billion in 2019 and is projected to reach nearly USD 24 billion by 2027 [30].



Taking into account the upward trend in the value of the gluten-free products market, consumer interest and the increasing availability and variety of these products, it is justified to undertake research aimed at determining the impact of the addition of chestnut flour on the texture of bread baked from gluten-free bread concentrates.

The novelty of the provided study is to provide the gluten-free products that can be addressed to the gluten-free community taking into consideration the storage time as well as the way of packaging.

## 2. Materials and Methods

Gluten-free bread concentrates were used with following ingredients:

- corn starch, Bogutyn Młyn, Poland;
- potato starch, Melvit, Poland;
- corn flour, Kupiec, Poland;
- chestnut flour, ViVio, Poland;
- instant yeast, Lesaffre, Poland;
- sugar, Suedzucker Polska Sp. z o.o, Poland;
- salt, Kłodawa S.A. Poland;
- hydrocypropyl methylcellulose (HPMC), J. Rettenmaier & Söhne, Poland

The following ingredients were added to the concentrates:

- milk, "Łaciate", Mlekpol, Poland, fat content 3.2%;
- potable water.

The composition of the chestnut flour:

- nutritional value w 100 g;
- energy value 1563 kJ/371 kcal—19% (Nutrient Reference Values);
- Fat 3.4 g—5% (saturated fatty acids 0 g—0%);
- carbohydrates 70 g—27% (sugar 18 g—20%);
- protein 7 g—14%;
- dietary fiber 16 g;
- salt 0.03 g < 1%.

### 2.1. Preparation of Gluten-Free Bread

The basis of gluten-free bread concentrates was corn starch. The same amount of potato starch, corn flour, instant yeast, sugar, salt and hydroxypropyl methylcellulose (HPMC) was present in each of the sample. Only the amount of corn starch and chestnut flour were changed. The concentrate recipes were established based on our preliminary findings, in which the amount of ingredients needed to prepare the dough was determined. The loose ingredients, which were used to prepare the mixture, were weighed in accordance with the recipes on technical scales with a precision of 0.01. The recipe composition is given in Table 1.

**Table 1.** The recipe composition of gluten-free bread concentrates.

| Ingredients | Raw Material Content [%] | | | | |
|---|---|---|---|---|---|
| Corn starch | 63.0 | 58.0 | 53.0 | 48.0 | 43.0 |
| Potato starch | | | 19.0 | | |
| Corn flour | | | 7.0 | | |
| Instant yeast | | | 2.4 | | |
| Sugar | | | 5.1 | | |
| Salt | | | 1.5 | | |
| HPMC | | | 2.0 | | |
| Chestnut flour | 0.0 | 5.0 | 10.0 | 15.0 | 20.0 |

All ingredients were mixed in mixer for 5 min; next, the mixture was stored in a plastic bowl for 30 min at a temp. of 40 °C. After 30 min, the dough was placed in the shaped bowls, where the process of fermentation was continued for the next 10 min until the

optimum volume was reached. Baking proceeded in the combi-steamer oven by UNOX (type XBC, model XBC 404) at a temp. of 175 °C for 23 min on the third level of vaporization. The refrigerated breads were packed using the packing machine EMPRA (VP 370) in a packaging made of PA/PE barrier foil (multilayer vacuum films laminated with polyamide) with air and vacuum (58%), and they were stored in room temperature (22 ± 2 °C). The temperature was set and controlled using air conditioning infrastructure. All measurements were made in 2 repetitions.

### 2.2. Determination of Volume, Moisture and Porosity

Bread volume was measured by the rapeseed displacement method 10-05. Bread moisture was determined according to the approved method 44-15A [31]. The porosity of the crumb was assessed by the differences between the volume of a bread crumb cylinder and the volume of a compressed crumb cylinder measured by oil displacement with a graduated cylinder. Moreover, the H/D ratio was determined (height/diameter).

### 2.3. Determination of Water Activity

The water activity determination was carried out on a Pre AquaLab Water Activity Analyzer (METER Group, Pullman, WA, USA). The samples were placed in airtight water activity cups and then placed in the cells of the apparatus. The measurements were made at the temperature of 20 ± 1 °C. After placing the samples in the apparatus, we waited about 3–5 min until the measured value stabilized. Measurements were made in five parallel replications. The final result was considered as the arithmetic mean of the measurements, after excluding the results differing from the others by 5%, considered as an error of the apparatus. In the tested products, both the initial water activity at zero point (baking time) and the water activity after 7, 14 and 21 days of storage were determined.

### 2.4. Determination of Crumb Hardness

Measurements of crumb hardness were done by using texture Analyzer TA-XTplus (Stable Micro Systems, Godalming, UK). The slices with a thickness of 20 mm were cut from the center of the analyzed loafs. Next, the slices were squeezed and relaxed. As before, we used a cylindrical head with an attachment in the shape of a cylinder of a diameter of 36 mm. Measurements were done at the speed of movement of the head of 1 mm/s, penetrating the sample to a depth of 10 mm with a charge cell of 250 N. The analysis of the moisture and the hardness of the crumb was made after 24 and 48 h after baking.

### 2.5. Sensory Analysis

Sensory analysis was carried out by 10 well-trained panelists. A nine-point hedonic scale was used to evaluate the overall acceptability of the bread formulations, ranging from 1 (dislike extremely) to 9 (like extremely) [32]. Samples of the same size were prepared for each panelist. Each assessor received bread samples in an identical form, which were marked in a coded manner and impossible to identify by the panelists.

### 2.6. Determination of Color (L*, a*, b*)

Characterization of the bread color was performed using the L*a*b* system proposed by the International Commission on Illumination (CIE) in the work of Papadakis [33]. L* refers to the luminosity or lightness component, and a* (intensity of red (+) and green (−)) and b* (intensity of yellow (+) and blue (−)) are the chromaticity coordinates. All sampled breads were analyzed in terms of the referred parameters using a Minolta CR-310 colorimeter (Konica-Minolta, Osaka, Japan), which was calibrated a priori with a white standard tile. Five repetitions were performed for each measurement.

### 2.7. Microbiological Quality

In the study, the microbiological quality of developed bread samples was investigated. Samples were evaluated immediately after production (zero point) and after 7, 14 and

21 days of storage at room temperature ($22 \pm 2°$ C) with lightening. Storage trials were carried out for 21 days (based on preliminary studies), because after the mentioned period, microbiological changes were observed, and the stored samples did not contain any added preservatives. Briefly, bread samples (10 g) were transferred to 90 mL peptone water (Lab M, Heywood, UK) and homogenized, serially diluted in sterile peptone water and surface spread on duplicate plates with the appropriate medium.

Nutrient agar (Biokar Diagnostic, Noack, Poland) was used for the enumeration of total viable counts (TVC), while MRS agar (LabM, Heywood, UK) was used for the enumeration of lactic acid bacteria (LAB). The plates were incubated at 30 °C for 72 h [34]. Chloramphenicol glucose agar (Biokar Diagnostic, Noack, Poland) and incubation at 25 °C for 120 h were used for enumeration of yeast and molds [35,36].

*Bacillus* spp. was investigated on agar PEMBA (LabM, Heywood, UK) [37].

### 2.8. Statistical Analysis

The statistical analysis of the obtained results was performed applying Statgraphics Plus 4.1., and the differences between the averages were estimated using multivariate regression analysis. The significance level ($\alpha$) was set to 0.05 and the smallest statistically significant difference was chosen using Tukey's test. PCA analysis was done using the software Statistica 13.3. The analyzed features were done in 3 repetitions.

## 3. Results and Discussion

### 3.1. Physical Properties

The breads with the chestnut flour in the amounts of 5, 10, 15 and 20% were analyzed, as well as the control sample, which was bread without the addition of chestnut flour. By analyzing the obtained results of the average weight value after baking (Table 2), it was found that the lowest weight characterized bread with a 10% addition of chestnut flour (153.88 g) followed by the control sample (160.97 g). No statistically significant differences were noticed in the case of the control sample and the bread with the 5% addition of chestnut flour. Moreover, no statistically significant differences were found for the weight of the bread with the 15% addition of chestnut flour and the weight of bread with the 20% addition of chestnut flour (166.69 g), which was characterized by the highest weight among all variants.

**Table 2.** Weight, volume, specific weight and porosity of gluten-free bread with the addition of chestnut flour in the amount of 0, 5, 10, 15 and 20%.

| Sample | Finished Product Weight [g] | Volume [cm$^3$/100 g] | Specific Mass [g/cm$^3$] | Porosity [%] |
|---|---|---|---|---|
| Control | $160.97 \pm 1.64$ b | $207.00 \pm 7.35$ a | $0.27 \pm 0.02$ c | $73.46 \pm 1.02$ c |
| 5% | $162.11 \pm 2.13$ b | $283.8 \pm 8.47$ b | $0.20 \pm 0.03$ b | $69.75 \pm 1.07$ b |
| 10% | $153.88 \pm 1.63$ a | $310.4 \pm 6.69$ b | $0.17 \pm 0.03$ a | $74.07 \pm 1.00$ c |
| 15% | $165.73 \pm 1.78$ c | $215.22 \pm 7.41$ a | $0.20 \pm 0.02$ b | $72.22 \pm 1.65$ c |
| 20% | $166.69 \pm 1.65$ c | $224.00 \pm 5.65$ a | $0.25 \pm 0.03$ c | $64.81 \pm 1.83$ a |

Values in the same column marked with the same symbols mean no statistically significant differences ($\alpha$ = 0.05).

The highest volume (310.4 cm$^3$/100 g) among the gluten-free breads was characteristic for the bread with the 10% addition of chestnut flour (Table 2). No statistically significant differences were found between the bread with the 10% addition of chestnut flour and the bread with the 5% addition of chestnut flour (283.8 cm$^3$/100 g). Comparing the variant without the addition of chestnut flour, with the breads with the 15% and 20% flour additions, no statistically significant differences were found in the values of this parameter. The value of this characteristic for breads with a variable addition of chestnut flour ranged from 207 cm$^3$/100 g to 310.4 cm$^3$/100 g. According to the standard for gluten-free bread [38], the volume of 100 g of bread should be no less than 190 cc. All the values obtained in the experiment are within the optimum range. Taking into account the obtained results, it was found that the addition of 5 and 10% chestnut flour to the recipe causes an increase in the

volume of the tested breads. Increasing the percentage of the addition of this flour causes a decrease in the baking volume. Similar relationships were obtained by Aguilar [39] in a study of the effect of chestnut flour leaven on the properties of gluten-free bread. The breads contained 15, 20 and 25% chestnut flour, and the basic raw material was corn starch. The increase in the concentration of chestnut flour caused a decrease in the volume of the bread. The research conducted by Demirkesen [40] on an attempt to replace rice flour with chestnut flour in bread showed that the volume of the bread increases with increasing the ratio of chestnut flour to rice flour. These differences may be due to the use of flour with different dietary fiber content. In this study, flour with the amount of fiber of 16% was used, whereas the amount of fiber in the flour used by Aguilar [39] was 15%, but Demirkesen [40] used chestnut flour with a fiber content of 9.5%. Due to its gas retention and viscoelastic properties, fiber can increase the volume of gluten-free bread. Too much fiber reduces the volume of the bread. These differences may also be caused by the use of various types of technological additives, e.g., emulsifiers [39].

The highest specific mass (Table 1) was found in the control sample (0.267 $g/cm^3$), the second in terms of this parameter was the bread with the 20% addition of chestnut flour (0.252 $g/cm^3$). The values of the specific weight of the breads with the 5% (0.197 $g/cm^3$) and 15% additions of chestnut flour (0.198 $g/cm^3$) did not differ statistically significantly. The performed statistical analysis showed a significant influence of the recipe composition on the specific weight of the tested bread. The largest statistically significant difference occurred between the control sample and the bread with a 10% addition of chestnut flour, which was characterized by the lowest value of the specific weight (0.174 $g/cm^3$).

The porosity is the ratio of the volume occupied by the pores to the total volume of the bread. High-quality bread is distinguished by a crumb with thin-walled and evenly spaced pores. For wheat bread, the porosity should be from 73 to 83%, and for rye bread, from 55 to 70% [41]. The lowest crumb porosity was characteristic for bread with the 20% addition of chestnut flour (64.8%), which was also characterized by the lowest volume. No statistically significant differences were found between the control sample (73.5%) and the bread with the 10% addition of chestnut flour (74.1%). These breads were characterized by the highest porosity among all the tested variants. Taking into account the obtained results and comparing them with the requirements concerning the porosity of the crumb of wheat bread, it was found that, apart from the control sample, the bread with the 10% addition of chestnut flour met the requirements specified in the standards for traditional bread. Similar results were obtained by Marciniak-Łukasiak and Skrzypacz [42], who investigated the effect of the addition of amaranth flour on the physicochemical and sensory properties of gluten-free bread. Amaranth flour was added (partially replacing the corn flour) in the amounts of 2.5, 5, 7.5, 10 and 12.5% of the total weight of the concentrate. The authors observed that the breads with the addition of amaranth flour in the amount of 5, 7.5 and 10% showed the highest porosity among all those baked in the series. Amaranth flour, added to the recipe in the amount of 12.5%, reduced the porosity of the gluten-free bread.

*3.2. Color*

One of the most important parameters in assessing the quality of food products and raw materials is color. The degree of the color intensity affects the positive or negative attitudes of consumers to a given product.

The L* parameter values for gluten-free breads ranged from 84.73 to 94.63 (Table 3). The highest value of this parameter was found in the control sample (94.63). Along with increasing the amount of the addition of chestnut flour, the value of the L* parameter decreased. No statistically significant differences were found between the variant of the breads with the 15% and 20% additions of this flour. These results are consistent with the results obtained by Aguliar [39], who, in the study of the properties of gluten-free bread, replaced rice flour with chestnut flour (15, 20, 25% chestnut flour). The L* brightness parameter showed that the greater the proportion of chestnut flour in the recipe, the darker

the color of the crumb. Similar observations were made by Rinaldi [43] when introducing chestnut flour to the bread recipe at the level of 20% (in relation to the amount of wheat flour). The addition of chestnut flour resulted in lower L* values (69.0 for bread made from wheat flour; 61.7 for bread with a 20% addition of chestnut flour). According to the authors, the cause of this phenomenon is not only the darker color of the raw material, but also the intense caramelization and Maillard reactions taking place due to the high sugar content in chestnut flour.

**Table 3.** The values of the color parameters of the crumb of gluten-free bread with the addition of chestnut flour in the amounts of 0, 5, 10, 15 and 20%.

| Sample | L* | a* | b* | Browning Index |
|---|---|---|---|---|
| Control | 94.63 ± 0.85 d | −0.22 ± 0.07 a | 14.54 ± 0.70 b | 16.31 ± 0.80 a,b |
| 5% | 90.37 ± 1.61 c | 0.11 ± 0.03 b | 12.65 ± 0.76 a | 15.01 ± 0.96 a |
| 10% | 87.77 ± 2.13 b | 0.23 ± 0.05 c | 12.86 ± 1.04 a | 16.42 ± 1.59 b |
| 15% | 85.08 ± 1.93 a | 0.34 ± 0.03 d | 14.73 ± 0.67 b,c | 18.45 ± 1.16 c |
| 20% | 84.73 ± 0.97 a | 0.45 ± 0.02 e | 15.54 ± 0.31 c | 20.40 ± 0.51 d |

Values in the same column marked with the same symbols (a–e) mean no statistically significant differences ($\alpha = 0.05$).

The value of the color chromaticity index (a*) of the crumb ranged from −0.22 (for the control sample) to 0.45 (for bread with the 20% addition of chestnut flour) and was statistically significantly differentiated (Table 3). The control sample was characterized by the predominance of a green color, while the highest share of a red color was the bread with the 20% addition of chestnut flour. Along with increasing the amount of the addition of chestnut flour, the value of the a* parameter increased. The analysis of the a* parameter value showed that higher amount of chestnut flour in the dough causes a larger proportion of the red color. In gluten-free breads studied by Aguilar [39], this tendency was also noticed; however, the values of this parameter were much higher, as they amounted to 4.87, 5.82 and 6.58, respectively, for bread with 15, 20 and 25% addition of chestnut flour, which may result from a different recipe composition.

The values of the b* parameter of gluten-free bread crumb ranged from 12.65 (for bread with a 5% addition of chestnut flour) to 15.55 (for bread with a 20% addition of chestnut flour) (Table 3). The lowest share of the shade of yellow in the color of the crumb was found in bread with the 5% addition of chestnut flour (12.65), and no statistically significant differences were found between bread with the 5% and 10% addition of chestnut flour (12.86). The b* parameter values for the control sample and the bread with the 15% addition of chestnut flour were 14.54 and 14.73, respectively. Comparing the obtained values, it was found that, with the increase in the concentration of chestnut flour, the share of the yellow color in the tested breads increases, which was also observed in [39], where the values of the b* parameter were 17.76, 18.47 and 19.14 for loaves with the 15, 20 and 25% addition of chestnut flour, respectively.

### 3.3. Browning Index

One of the parameters that proves the color change is the browning index (BI). It represents the purity of a brown color and is reported as an important parameter in processes where enzymatic and nonenzymatic browning occurs [44].

The obtained browning index values for gluten-free bread are presented in Table 3 and were statistically significantly differentiated. The mean value of the browning index for the control sample was 16.31. With the addition of chestnut flour to the recipe, it was observed that the browning index values gradually increase. However, the browning index value for bread with the 5% addition of chestnut flour is lower compared to the control sample.

### 3.4. Moisture

Moisture is one of the most important physical and chemical parameters of the bread crumb. It mainly shows the degree of the freshness of the bread and influences the staling process, which adversely affects changes in the sensory characteristics of the bread [45].

The purpose of determining the humidity of the crumb of gluten-free bread at point zero after 7, 14 and 21 days of storage was to compare the changes occurring during their storage. After 7, 14 and 21 days, the samples were stored and packed in two ways: in a barrier foil with air access and in a barrier foil with a vacuum (58%) at a temperature of $22 \pm 2$ °C.

According to the standard [38], the humidity of gluten-free bread should not exceed 53%. The tested gluten-free breads packed in a barrier foil with air had a humidity ranging from 38.89% to 47.26%, so it was within the norm (Figure 1). Research conducted by Cacak-Pietrzak [20] also confirms that the humidity of gluten-free bread ranges from 34.3% to 49.7%. The control sample (47.26%) was characterized by the highest humidity of the crumb at the zero point. With the increase of the storage time, the humidity of the crumb in the control sample gradually decreased to the level of 42.03%. All breads had the highest humidity at the zero point. The obtained relationships are consistent with the observations of Demirkesen [46], who examined the moisture content of gluten-free bread baked with rice flour and the addition of chestnut flour. They showed that, during the storage of bread, the process of staling occurs due to the migration of moisture from the crumb to the crust. As a result of this process, the ability to bind water by the crumb decreases, so the lowest moisture losses in the bread are observed in the samples stored for the shortest time. In the case of breads with the addition of chestnut flour, a decrease in humidity was observed after 7 days of storage. The lowest humidity was observed after 21 days of storage in all variants of gluten-free bread, except for bread with the 5% addition of chestnut flour, the humidity of which was 39.10%.

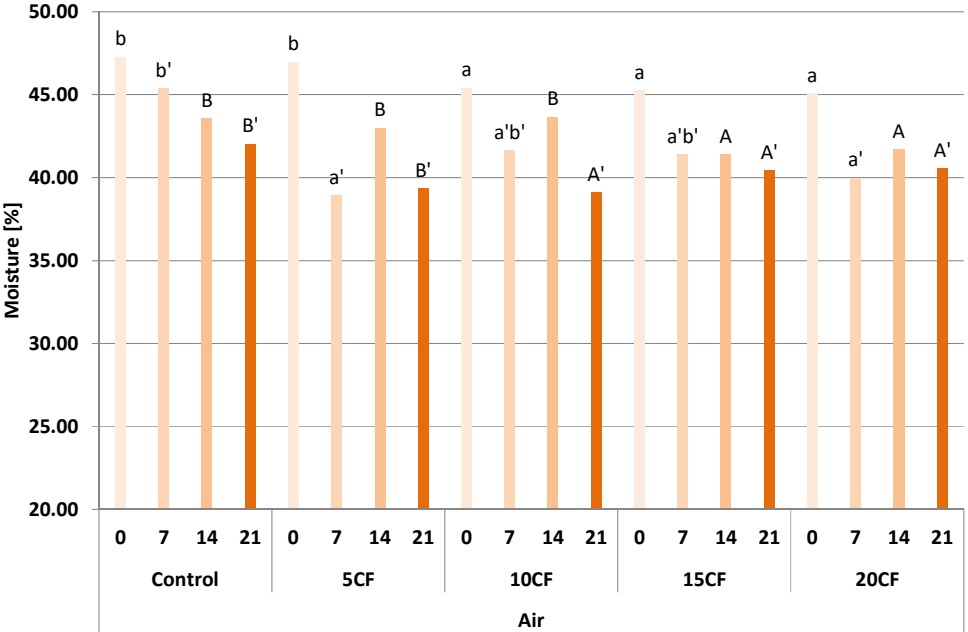

**Figure 1.** Moisture of the crumb of gluten-free breads with the addition of chestnut flour in the amount of 5% (5CF), 10% (10CF), 15% (15CF) and 20% (20CF) after 7, 14 and 21 days of storage, packed in a barrier film with air. Values marked with the same symbols mean no statistically significant differences ($\alpha = 0.05$).

The moisture content of the vacuum-packed gluten-free bread ranges from 38.56% to 47.26%, so it complies with the standard [38] (Figure 2). As the bread storage time increased,

the crumb moisture decreased successively, except for the breads with the 5% and 10% addition of chestnut flour. In both cases, the humidity increased on the 21st day of storage.

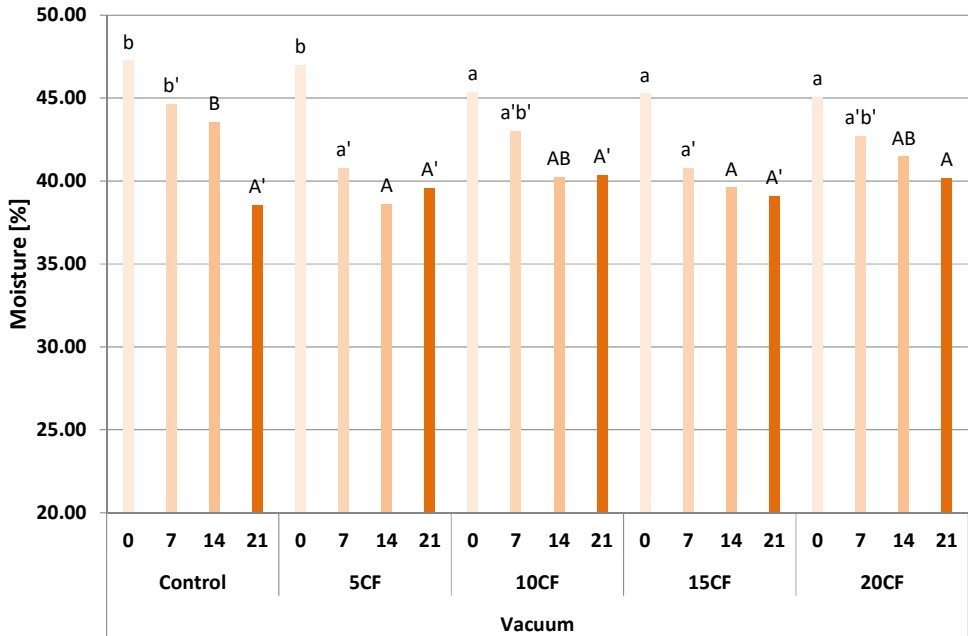

**Figure 2.** Moisture of the crumb of gluten-free breads with the addition of chestnut flour in the amount of 5% (5CF), 10% (10CF), 15% (15CF) and 20% (20CF) after 7, 14 and 21 days of storage, packed in a barrier film with a vacuum. Values marked with the same symbols mean no statistically significant differences (α = 0.05).

Comparing the method of packing the tested breads, higher humidity was observed in the case of the breads with the 5, 10 and 20% addition of chestnut flour, vacuum-packed, after 7 days of storage. The control sample and the bread with 15% addition of chestnut flour were characterized by lower humidity compared to the bread packed with air access. After 14 days of storage, the humidity of the crumb was lower in the vacuum-packed bread, with the exception of the control sample, in which the humidity of the bread packed with air was 43.57%, while in vacuum-packed bread it was 43.58%. However, these differences are not statistically significant. After 21 days of storage, in the control and the breads with the 15 and 20% addition of chestnut flour lower humidity in the vacuum-packed samples was observed.

### 3.5. Water Activity

On the basis of the conducted research, it was found that the fresh bread at the zero point was characterized by a higher water activity compared to the bread tested after the given storage time. Water activity in the breads with the 5% and 20% addition of chestnut flour gradually decreased until they were stored. In the case of the control sample, the bread with the 15% and the bread with the 20% addition of chestnut flour, water activity increased after 14 days compared to its level after 7 days of storage. It can be assumed that the reason for this is the separation of free water in the product, which contributed to its evaporation into the environment [38]. The water activity in the analyzed breads packed in a barrier foil with air ranges from 0.956 to 0.983 (Figure 3).

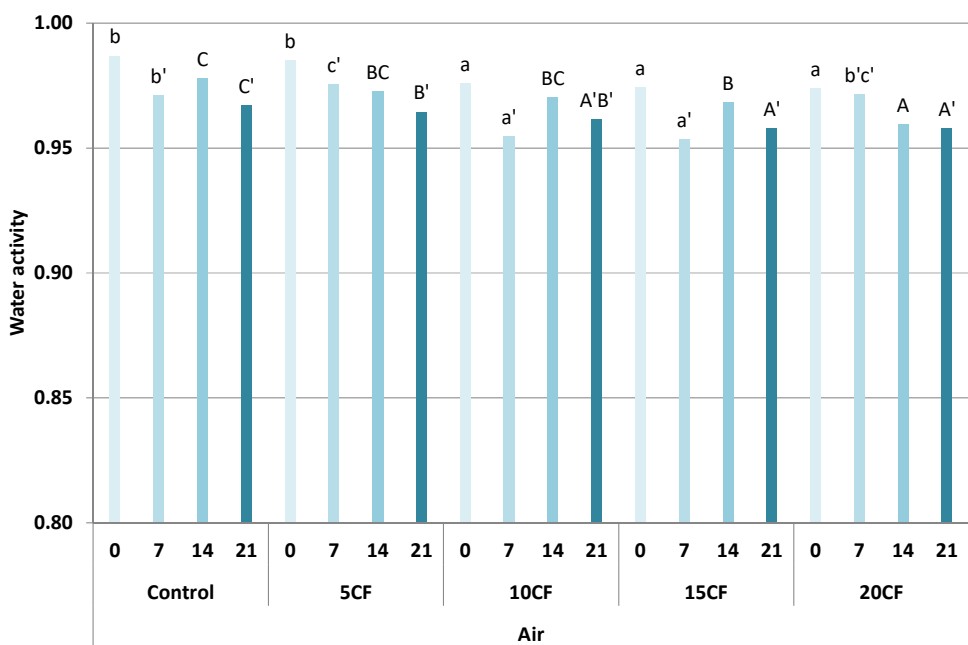

**Figure 3.** Water activity of gluten-free breads with the addition of chestnut flour in the amount of 5% (5CF), 10% (10CF), 15% (15CF) and 20% (20CF) after 7, 14 and 21 days of storage, packed in a barrier film with air. Values marked with the same symbols mean no statistically significant differences ($\alpha = 0.05$).

These results are on a similar level to the results obtained by Pałacha and Makarewicz [47], who examined the crumb of various types of bread, the water activity of which ranged from 0.926 to 0.975. Similar results were obtained by Aguilar [39], who tested the water activity in gluten-free bread with the addition of 15, 20 and 25% chestnut flour at the zero point and after 7 days of storage. It was shown that the water activity after 7 days of storage decreased compared to the water activity at the zero point. The water activity of the breads with 15, 20 and 25% chestnut flour added at the zero point was 0.975, respectively, 0.975 and 0.973, and after 7 days 0.972, 0.970 and 0.970. As in the case of air-packed bread, the water activity was highest in fresh bread—at the zero point. Water activity in the control sample and the breads with 5 and 15% addition of chestnut flour increased after 14 days compared to its level after 7 days of storage, and the breads with the 10 and 20% addition of chestnut flour was characterized by higher water activity after 21 days after baking compared to its level after 14 days of storage. The range of water activity in the analyzed vacuum-packed breads ranged from 0.949 to 0.983 (Figure 4). The factors that prolong the shelf life of products and ensure their microbiological stability, despite the high activity of water, could be contributed by: high baking temperature and the presence of a crust on the surface of the bread, which is a natural protection against external factors [47].

Comparing the method of packing the tested gluten-free breads, higher water activity was observed in the control sample and in the vacuum-packed bread with the 10% addition of chestnut flour after 7 days of storage. After 14 days of storage, the water activity was lower in the vacuum-packed bread, except for the control sample, in which the water activity of the air-packed bread was 0.974, and in the vacuum was 0.978. After 21 days of storage in the control sample and the breads with the 5% and 15% addition of chestnut flour, lower water activity was observed in the samples packed under vacuum.

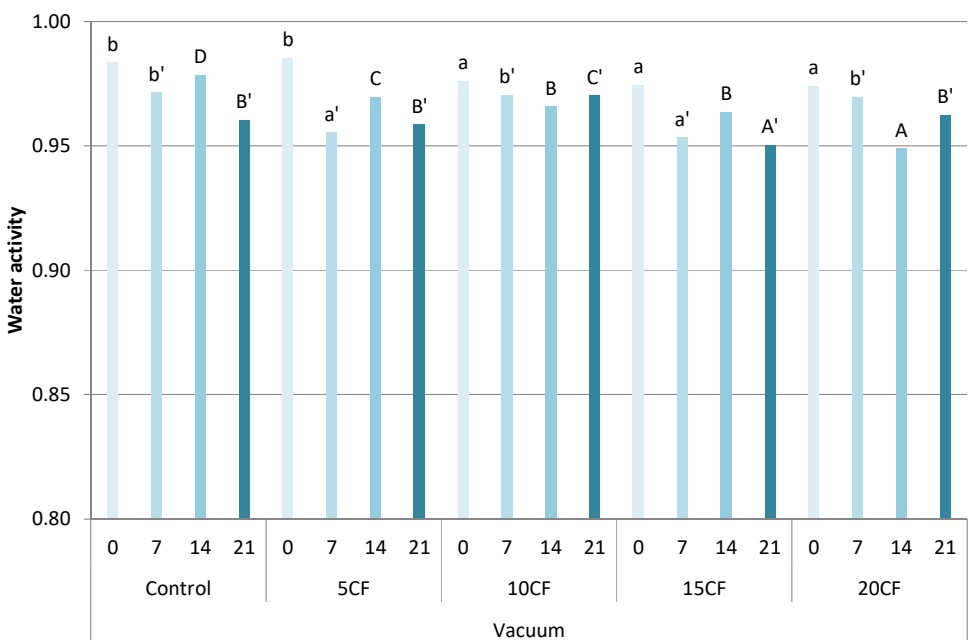

**Figure 4.** Water activity of gluten-free breads with the addition of chestnut flour in the amount of 5% (5CF), 10% (10CF), 15% (15CF) and 20% (20CF) after 7, 14 and 21 days of storage, packed in a barrier film with a vacuum. Values marked with the same symbols mean no statistically significant differences ($\alpha = 0.05$).

*3.6. Texture*

3.6.1. Hardness

Hardness is defined as the peak force during the first compression cycle [48].

The crumb hardness of gluten-free breads packed in a zero point with an air barrier film was from 1.25 N (for the bread with the 5% addition of chestnut flour) to 7.32 N (for the control) (Table 4). No statistically significant differences were found between the variants with the 10, 15 and 20% addition of chestnut flour. The addition of chestnut flour significantly reduced the hardness of the crumb. The 5% share of the chestnut flour in the tested breads turned out to be the most advantageous in this respect. An increase in the crumb hardness was observed after 7 days of storage in all analyzed samples. The increase in bread hardness results from its staleness. The main reason for this process is the transformation of starch from its amorphous to pseudocrystalline form (starch retrogradation). This form binds smaller amounts of water, which increases the brittleness and hardness of the crumb [49,50]. After 14 days of storage, an increase in the crumb hardness was observed only in the case of bread with the 20% addition of chestnut flour. Twenty-one days after baking, a significant increase in crumb hardness was observed in the control sample (18.87 N), while the breads with the 5, 10 and 15% addition of chestnut flour were characterized by lower hardness values compared to the crumb hardness on the 14th day of storage.

As in the case of the samples packed with air, an increase in the crumb hardness of the gluten-free bread packed in a barrier foil with a vacuum was observed after 7 days of storage in all analyzed variants (Table 4). After 14 days of storage, an increase in the crumb hardness was noticed only in the case of the breads with the 15 and 20% addition of chestnut flour. Twenty-one days after baking, the crumb hardness increased in the control sample and the breads with the 5% and 10% addition of chestnut flour. Gambuś [51] investigated the effect of an amaranth flour addition on the quality of gluten-free bread. The authors observed an increase in the crumb hardness with an increasing storage time. Similar results were also obtained by Kulczak [52], who assessed selected physical properties of gluten-free bread with the use of instant pea flour and buckwheat products. The crumb hardness

of the tested samples was assessed 3 h, 24 h and 48 h after baking. An increase in the crumb hardness was observed with an increasing storage time. These results are partially consistent with the results presented in this study, because, both in the case of bread packed with air and in vacuum, an increase in the hardness of the crumb of the tested bread was observed after 7 days of storage.

**Table 4.** Hardness, elasticity, cohesiveness, gumminess and chewiness of gluten-free bread with the addition of chestnut flour in the amounts of 0, 5, 10, 15 and 20%.

|  | Sample | Time [day] | Hardness [N] | Elasticity | Cohesiveness | Gumminess | Chewiness [N] |
|---|---|---|---|---|---|---|---|
| Air | Control | 0 | 7.32 c | 2.37 a,b | 0.41 a,b | 2.83 b | 7.89 b |
|  |  | 7 | 11.19 b′ | 4.81 a | 0.51 a′ | 5.67 a′ | 27.32 a′ |
|  |  | 14 | 10.82 E | 1.37 A | 0.29 A | 3.26 A | 7.25 A |
|  |  | 21 | 18.87 D′ | 1.70 A′ | 0.58 B′ | 10.90 C′ | 17.32 A′,B′ |
|  | 5% | 0 | 1.25 a | 0.98 a | 0.17 a | 0.25 a | 1.21 a |
|  |  | 7 | 7.46 a′ | 4.94 a′ | 0.73 a′,b′ | 5.48 a′ | 27.06 a′ |
|  |  | 14 | 4.14 A | 3.67 C | 0.72 B | 2.96 A | 11.12 A,B |
|  |  | 21 | 4.04 A′ | 1.63 A′ | 0.28 A′ | 1.11 A′ | 5.42 A′ |
|  | 10% | 0 | 3.21 b | 2.13 a,b | 0.46 a,b | 1.59 a,b | 6.24 b |
|  |  | 7 | 7.02 a′ | 4.59 a′ | 0.82 a′,b′ | 5.78 a′ | 26.53 a′ |
|  |  | 14 | 5.86 B | 1.70 A | 0.41 A | 2.43 A | 7.74 A |
|  |  | 21 | 4.02 A′ | 4.88 B′ | 0.91 D′ | 3.67 A′,B′ | 17.92 A′,B′ |
|  | 15% | 0 | 2.43 b | 3.39 a,b | 0.64 a,b | 1.53 a,b | 6.97 b |
|  |  | 7 | 7.49 a′ | 4.31 a′ | 0.74 a′,b′ | 5.40 a′ | 23.29 a′ |
|  |  | 14 | 6.55 C | 2.52 B | 0.59 A,B | 3.92 A | 13.75 A,B |
|  |  | 21 | 6.49 B′ | 4.70 B′ | 0.76 C′ | 4.88 A′,B′ | 23.06 B′ |
|  | 20% | 0 | 2.55 b | 4.45 b | 0.83 b | 2.13 a,b | 9.75 b |
|  |  | 7 | 9.46 b′ | 4.90 a′ | 0.92 b′ | 8.70 b′ | 42.63 b′ |
|  |  | 14 | 9.63 D | 3.21 C | 0.81 B | 7.82 B | 26.06 B |
|  |  | 21 | 10.00 C′ | 1.73 A′ | 0.59 B′ | 5.89 B′ | 11.48 A′ |
| Vacuum | Control | 0 | 7.32 c | 2.37 a,b | 0.41 a,b | 2.83 b | 7.89 b |
|  |  | 7 | 7.63 c′ | 2.04 a′ | 0.61 a′,b′ | 4.61 b′ | 10.12 a′,b′ |
|  |  | 14 | 3.92 A | 3.60 C | 0.72 B | 2.80 A,B | 9.46 A |
|  |  | 21 | 18.63 D′ | 1.00 A′ | 0.29 A′ | 5.34 A′,B′ | 8.31 A′ |
|  | 5% | 0 | 1.25 a | 0.98 a | 0.17 a | 2.50 a | 1.21 a |
|  |  | 7 | 4.42 a′ | 3.81 b′ | 0.77 b′ | 3.41 a′,b′ | 14.10 a,b |
|  |  | 14 | 4.31 A | 1.66 A | 0.27 A | 1.12 A | 5.57 A |
|  |  | 21 | 5.25 A′ | 3.02 B′ | 0.77 C′ | 4.07 A′ | 12.83 A′ |
|  | 10% | 0 | 3.21 b | 2.13 a,b | 0.46 a,b | 1.59 a,b | 6.24 b |
|  |  | 7 | 4.68 a′,b′ | 1.49 a′ | 0.34 a′ | 1.54 a′ | 4.96 a′ |
|  |  | 14 | 3.55 A | 2.61 B | 0.68 B | 2.39 A | 6.53 A |
|  |  | 21 | 6.55 B′ | 4.10 C′ | 0.64 B′ | 4.18 A′ | 17.11 A′ |
|  | 15% | 0 | 2.43 b | 3.39 a,b | 0.64 a,b | 1.53 a,b | 6.97 b |
|  |  | 7 | 5.48 b′ | 4.02 b′ | 0.90 b′ | 4.88 b′ | 19.87 b′ |
|  |  | 14 | 7.40 B | 3.92 C | 0.76 B | 5.60 B | 23.38 A,B |
|  |  | 21 | 5.32 A′ | 4.51 D′ | 0.77 C′ | 4.04 A′ | 18.28 A′ |
|  | 20% | 0 | 2.55 b | 4.45 b | 0.83 b | 2.13 a,b | 9.75 b |
|  |  | 7 | 10.00 d′ | 3.00 a′,b′ | 0.72 a′,b′ | 7.09 c′ | 21.23 b′ |
|  |  | 14 | 11.76 C | 4.43 D | 0.76 B | 8.99 C | 40.08 B |
|  |  | 21 | 10.69 C′ | 2.16 A′B′ | 0.67 B′ | 7.12 B′ | 14.67 A′ |

Values in the same column marked with the same symbols mean no statistically significant differences ($\alpha = 0.05$).

Comparing the method of storage of the tested variants, it was noticed that the samples packed in the vacuum (except for the bread with the 20% addition of chestnut flour) after 7 days were characterized by a lower crumb hardness compared to the samples packed with air. After 14 days from baking, the control sample and the vacuum-packed bread with

the 10% chestnut flour had a lower crumb hardness compared to the samples packed with air, and 21 days after baking, the vacuum-packed breads with 5, 10 and 20% chestnut flour had a lower crumb hardness. harder crumb than air-packed bread.

### 3.6.2. Elasticity

Elasticity is defined as the quotient of the specimen deformation that occurs during the first and second compression cycles. It characterizes the degree of the shape recovery by the sample [48].

The lowest crumb elasticity of gluten-free bread packed in a barrier foil with air at the zero point was found in bread with the 5% addition of chestnut flour (0.98) (Table 4). As the amount of the addition of chestnut flour in the recipe increased, the elasticity of the bread crumb increased. The highest value of elasticity was characteristic for the bread with the 20% addition of chestnut flour (4.45). No statistically significant differences were observed between the control sample and the breads with the 10% and 15% addition of chestnut flour.

Seven days after baking, crumb elasticity increased significantly in all tested variants, and no statistically significant differences were observed between the samples. After 14 days of storage, a significant decrease in the crumb elasticity was observed compared to the values measured on the 7th day after baking. The lowest values of elasticity were found in the control sample (1.37) and the bread with the 10% addition of chestnut flour (1.70). There were no statistically significant differences between the breads with the 5% and 20% addition of chestnut flour. After 21 days of storage, a decrease in crumb elasticity was noted in the case of bread with the 5% and 20% addition of chestnut flour. No statistically significant differences were observed between the control sample, the breads with 5% and 20% addition of chestnut flour and between the bread with the 10% and 15% addition of chestnut flour. Similar relationships were observed in the studies by Marciniak-Lukasiak and Skrzypacz [52], where the addition of amaranth flour was used.

After 7 days of storage, a decrease in the crumb elasticity of the gluten-free breads packed in a barrier foil with a vacuum was observed in the case of the control and the breads with the 10% and 20% addition of chestnut flour (Table 4). In the case of the breads with the 5% and 15% addition of chestnut flour, an increase in crumb elasticity was noted. After 14 days from baking, the crumb elasticity of the control sample and the breads with the 10% and 20% addition of chestnut flour increased in comparison to the values measured after 7 days of storage. No statistically significant differences were observed for the control sample and for the bread with the 10% addition of chestnut flour. After 21 days of storage, a significant decrease in the crumb elasticity was noted in the control sample and in the bread with the 20% addition of chestnut flour.

Comparing the method of storage of the tested variants, it was noticed that the vacuum-packed samples after 7 days of storage were characterized by lower crumb elasticity values compared to the samples packed with air. However, such a tendency was not noticed 14 and 21 days after baking. After 14 days of storage, a lower value of the crumb elasticity was observed only in the case of bread with the 5% addition of chestnut flour, vacuum-packed, and after 21 days from baking, while the control sample and the breads with the 10% and 15% addition of chestnut flour showed lower values of crumb elasticity compared to the samples packed with air.

### 3.6.3. Cohesiveness

Cohesiveness (cohesiveness) characterizes the total strength of the internal bonds that hold the product together [53].

The values of the crumb cohesiveness parameter of the gluten-free bread packed with air at the zero point ranged from 0.17 to 0.83 (Table 4). On the basis of the obtained results, it was found that with the increase in the amount of the addition of chestnut flour in the recipe, the cohesiveness increases. However, the cohesiveness of the bread with the 5% addition of chestnut flour (0.17) was lower compared to the control sample (0.41). There

were no statistically significant differences between the control sample and the breads with the 10% and 15% addition of chestnut flour.

After 7 days from baking, the value of the cohesiveness parameter increased. It was observed that, the larger the proportion of chestnut flour in the recipe, the smaller were the differences in the cohesiveness values. These results differ from the results obtained by Gambuś [51]. The authors observed in their research that the storage time causes a decrease in the cohesiveness. After 14 days from baking, there was a decrease in the cohesiveness in all the analyzed samples compared to their value after 7 days of storage. No statistically significant differences were observed between the control sample and the bread with the 10% addition of chestnut flour. Twenty-one days after baking, there was a decrease in cohesiveness in breads with the 5% and 20% addition of chestnut flour compared to their value after 14 days after baking.

In the case of the gluten-free breads packed in a barrier foil with a vacuum with the 10% and 20% addition of chestnut flour, lower cohesiveness values were obtained (0.34 for 10%; 0.72 for 20%) compared to the cohesiveness at the zero point (0.46 for 10%; 0.83 for 20%). The results obtained for these two variants confirm the observations made by Marciniak-Lukasiak and Skrzypacz [42], which, in their research, showed that with an increasing storage time, the cohesiveness decreases. However, it cannot be concluded in this study that the storage time caused a decrease in the cohesiveness. In the case of the control sample and the breads with the 5% and 15% addition of chestnut flour, an increase in the cohesiveness was observed 7 days after baking. After 14 days of storage, a decrease in the cohesion value for the breads with the 5 and 15% addition of chestnut flour was observed, while after 21 days of storage, a decrease was noted for the control sample and for the breads with the 10 and 20% addition of chestnut flour.

Comparing the method of storage of the tested variants, it was noticed that the vacuum-packed samples after 7 days of storage were characterized by lower crumb cohesiveness values compared to the samples packed with air. However, such a relationship was not observed with increasing the bread storage time. After 14 days from baking, the cohesiveness of the vacuum-packed variants was lower only in the case of the breads with the 5% and 20% addition of chestnut flour, and after 21 days of storage in the case of the control sample and the bread with the 10% addition of chestnut flour.

### 3.6.4. Gumminess

Gumminess is the quotient of the hardness and cohesion of the bread crumb [53].

The crumb chewiness of the gluten-free breads packed with air access at the zero point was from 0.25 to 2.83 (Table 4). Bread with the 5% addition of chestnut flour was characterized by the lowest gumminess, and the control sample was the highest. All variants with the addition of chestnut flour reduced the crumb gumminess compared to the control sample. No statistically significant differences were found between the gumminess of the bread crumb with the 10, 15 and 20% addition of chestnut flour.

After 7 days of storage, an increase in the gumminess was observed in all analyzed samples. No statistically significant differences were observed between the control sample and the breads with the 5, 10 and 15% addition of chestnut flour. Pajak [54] observed a decrease in the value of this parameter in the research on the impact of packaging on the quality of stored gluten-free bread. After 14 days from baking, there was a decrease in the gumminess in all analyzed samples compared to their values tested on the 7th day of storage. As in the samples tested after 7 days, no statistically significant differences were observed between the control sample and the breads with the 5, 10 and 15% addition of chestnut flour. On the 21st day after baking, a sharp increase in the gumminess was noted in the control sample (10.90). A decrease in the value of this parameter was observed in the case of the breads with the 5% and 20% addition of chestnut flour.

As with the air-wrapped bread, after 7 days of storage, the gumminess of the gluten-free bread wrapped in a barrier film with the vacuum increased, except for the bread with 10% chestnut flour. The zero point gumminess was 1.59, and after 7 days it was 1.54

(Table 4). Similar relationships were observed in the case of the cohesiveness value of this variant. After 14 days of storage, a decrease in gumminess was noted in the case of the control sample and the bread with the 5% addition of chestnut flour, while 21 days after baking, the gumminess of the control sample and the breads with the 5% and 10% addition of chestnut flour increased in comparison to the 14th day of storage.

Comparing the results obtained during the storage of the tested variants, it was noticed that the vacuum-packed samples, after 7 days of storage, were characterized by lower crumb gumminess values compared to the samples packed with air. After 14 days from baking, the gumminess of the vacuum-packed variants was lower in the control sample and the breads with the 5% and 20% addition of chestnut flour, and after 21 days of storage in the control sample and the bread with the 15% addition of chestnut flour.

### 3.6.5. Chewiness

Chewiness is the product of hardness, elasticity and cohesiveness. It characterizes the strength needed to chew a bite of food so that it is ready to be swallowed [53]

The chewiness of the crumb of gluten-free bread packed with air access at the zero point was from 1.21 N to 9.75 N (Table 4). On the basis of the obtained results, it was found that, with the increase in the amount of chestnut flour added to the recipe, the chewiness of the crumb grows. The lowest chewiness value was characteristic for the bread with the 5% addition of chestnut flour, and the highest value for the bread with the 20% addition of chestnut flour. The 5, 10 and 15% addition of chestnut flour in the gluten-free breads reduced the chewiness of their crumb. Pajak [54] observed in their research that the value of the chewing parameter of the crumb of gluten-free bread decreases during storage. After 14 days from baking, lower chewiness was noted in each of the tested variants. Both after 7 and 14 days of storage, the bread with the greatest addition of chestnut flour was characterized by the greatest chewiness of the crumb. After 21 days, a decrease in chewiness was observed only for the breads with the 5% and 20% addition of chestnut flour, as compared with its values on the 14th day after baking.

As with air-packed breads, after 7 days of storage, the chewiness of the gluten-free breads wrapped in a barrier film with a vacuum increased, with the exception of the bread with 10% chestnut flour (Table 4). The chewiness at the zero point was 6.24 N, and after 7 days, it was 4.96 N. Such relations were observed for the cohesiveness and gumminess values of this variant. After 14 days of storage, a decrease in the chewiness was noted for the control sample and the bread with the 5% addition of chestnut flour, while 21 days after baking, the chewiness of the breads with the 5% and 10% addition of chestnut flour increased compared to the 14th day of storage.

Comparing the method of storage of the tested variants, it was noticed that the vacuum-packed samples after 7 days of storage were characterized by lower crumb chewiness values compared to the samples packed with air. This relationship was not observed with an increasing bread storage time. After 14 days from baking, the chewiness of the vacuum-packed variants was lower in the case of the breads with the 5 and 10% addition of chestnut flour, and after 21 days of storage in the case of the control sample and the breads with the 10 and 15% addition of chestnut flour.

### 3.7. Sensoric Analysis

One of the discriminants subjected to the sensory assessment was taste (Figure 5). The assessed tastes are bready, salty, sour, yeast and foreign. A total of 100 UU was assumed as the highest intensity, and 0 for the lowest.

The bread with a 5% addition of chestnut flour (74.3 IU) was the most bready, i.e., the typical taste, and the second in terms of this feature was the bread with the 10% addition of chestnut flour (72.0 IU). The least bready taste (58.0), and at the same time having foreign aftertastes (13.8), which the evaluators described as slightly nutty, was the bread with the 20% addition of chestnut flour. According to the evaluators, all the breads with the addition of chestnut flour had a slightly more perceptible sour taste (4.0 IU) compared to

the control sample (3.7 IU). The amount of the addition of chestnut flour used did not have a statistically significant effect on the taste perception of the added yeast.

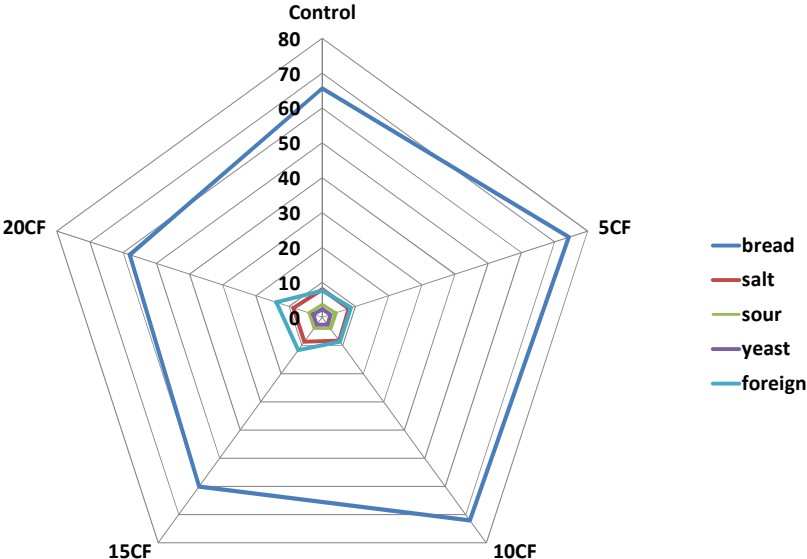

**Figure 5.** The taste of gluten-free bread with the addition of chestnut flour in the amount of 5% (5CF), 10% (10CF), 15% (15CF) and 20% (20CF).

Another distinguishing feature during the sensory evaluation was smell (Figure 6). The scents assessed were: bread-like, the smell of added yeast and the presence of foreign smells. In terms of the bread flavor, the best were the breads with the 5% and 10% addition of chestnut flour (76.0 u.u.). These breads were also rated the highest in terms of the typical bread flavor. Bread with a 20% addition of chestnut flour (58.3 u.u.) and the control sample (59.0 u.u.) had the least bread smell. According to the evaluators, these two variants were also characterized by an intense smell of added yeast (67.3 u.u. in the control sample and 63.7 u.u. in bread with 20% addition of chestnut flour). The presence of foreign smells was particularly significant in the case of the bread with the highest share of chestnut flour (45.3 u.u.), which the evaluators described as nutty, while in the control sample, the evaluators experienced slightly sour scent notes (16.0 u.u.).

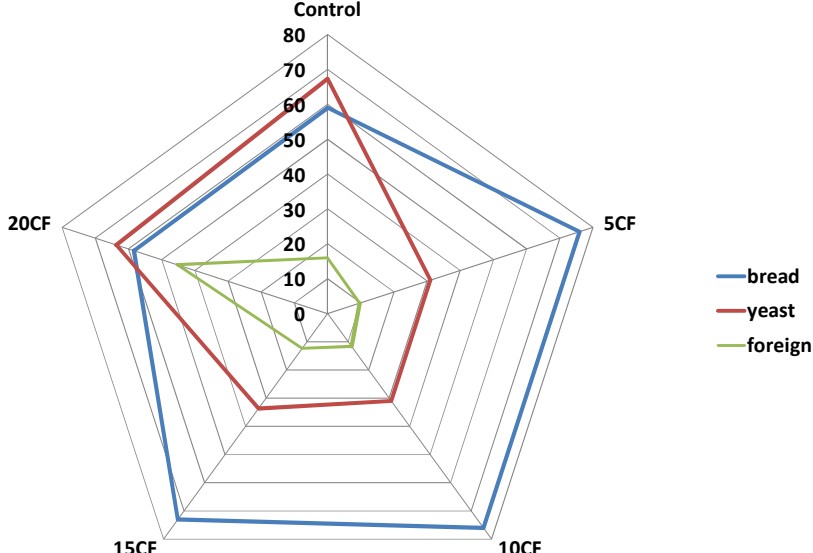

**Figure 6.** The smell of gluten-free bread with the addition of chestnut flour in the amount of 5% (5CF), 10% (10CF), 15% (15CF) and 20% (20CF).

The structure and texture of the crumb of the baked gluten-free bread were also assessed (Figure 7). The hardest (86.5 u.u.), and at the same time, the most fragile (81.3 u.u.) and the most compact (86.3 u.u.) crumb was characteristic of the control sample. The highest hardness of this bread was also confirmed in an instrumental TPA test. Similar values were obtained for the test with the 20% addition of chestnut flour—a hardness of 81.2 u.u., a brittleness of 73.2 u.u. and a compactness of 81.0 u.u. The lowest scores of all three features of the structure and texture were recorded for the bread with the 5% addition of chestnut flour (hardness 22.3 u.u.; brittleness 31.7 u.u.; compactness 23.3 u.u.), which was also confirmed in the TPA test. No significant differences were observed in the case of the bread with the 10% addition of chestnut flour, where the brittleness was assessed at the level of 33.3 u.u., the hardness of 23.0 u.u. and the compactness of 25.7 u.u.

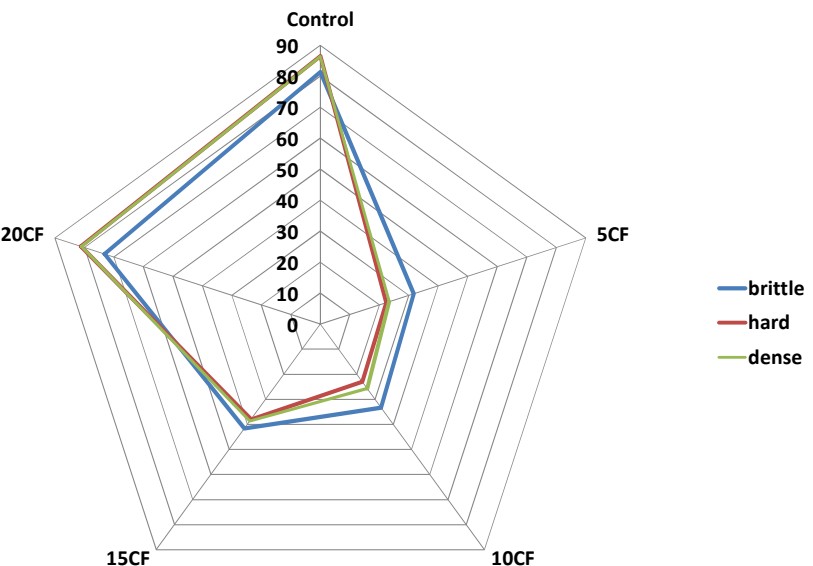

**Figure 7.** The structure and texture of the crumb of gluten-free breads with the addition of chestnut flour in the amount of 5% (5CF), 10% (10CF), 15% (15CF) and 20% (20CF).

The last of the assessed characteristics of baked gluten-free bread was the assessment of consumer desirability, which is presented in Figure 8.

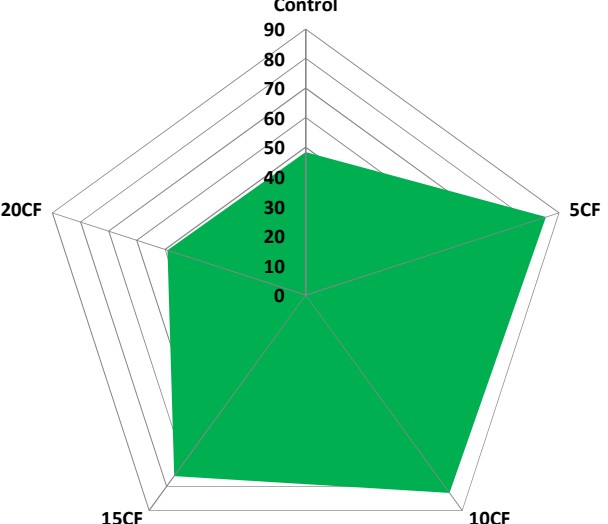

**Figure 8.** Consumer assessment for gluten-free breads with the addition of chestnut flour in the amount of 5% (5CF), 10% (10CF), 15% (15CF) and 20% (20CF).

Among the examined breads, the most desired by consumers was the bread with the 5% addition of chestnut flour (85.3 u.u.) and the bread with the 10% addition of chestnut flour (82.7 u.u.). The desirability of the bread with the 15% addition of chestnut flour was estimated at 75.7 u.u. In the evaluation of the gluten-free breads obtained, the control sample (48.3 u.u.) and the bread with the 20% addition of chestnut flour (49.2 u.u.) showed the lowest consumer desires.

### 3.8. Microbiological Quality

Microbiological quality of bread samples with the chestnut flour addition differed significantly depending on the method and the time of storage (Table 5). Immediately after baking, no microorganisms were detected in the bread samples; however, after 7, 14 and 21 days of storage, 3.47–5.87 log cfu/g of mesophilic aerobic microorganisms, as well as 2.11–5.47 log cfu/g of yeast and molds were denoted. Moreover, in the samples containing 20% of chestnut flour, *Bacillus* spp. was detected, but only after 7, 14 and 21 days of storage. Interestingly, the microbiological quality of the samples in vacuum storage was better than the aerobic conditions, which was seen in the TCV and Y&M counts.

**Table 5.** TVC, Y&M, LAB and *Bacillus* spp. of gluten-free bread with the addition of chestnut flour in the amount of 0, 5, 10, 15 and 20%.

| | Chestnut Flour Addition | TVC | | Y&M | | LAB | | *Bacillus* spp. | |
|---|---|---|---|---|---|---|---|---|---|
| | | Air | Vacuum | Air | Vacuum | Air | Vacuum | Air | Vacuum |
| 0 | 0% | <10 | <10 | <10 | <10 | <10 | <10 | Nd | nd |
| | 5% | <10 | <10 | <10 | <10 | <10 | <10 | Nd | nd |
| | 10% | <10 | <10 | <10 | <10 | <10 | <10 | Nd | nd |
| | 15% | <10 | <10 | <10 | <10 | <10 | <10 | Nd | nd |
| | 20% | <10 | <10 | <10 | <10 | <10 | <10 | Nd | nd |
| 7 | 0% | 3.53 ± 0.21 a | 2.98 ± 0.15 b | 2.11 ± 0.06 a | 1.60 ± 0.09 a | <10 | <10 | Nd | nd |
| | 5% | 3.59 ± 0.16 a | 2.47 ± 0.23 a | 2.92 ± 0.02 b | 2.17 ± 0.06 b | <10 | <10 | Nd | nd |
| | 10% | 3.48 ± 0.11 a | 2.76 ± 0.20 ab | 2.82 ± 0.08 b | 2.60 ± 0.09 c | <10 | <10 | Nd | nd |
| | 15% | 3.78 ± 0.14 a | 2.95 ± 0.14 b | 3.48 ± 0.13 c | 2.25 ± 0.15 b | <10 | <10 | Nd | nd |
| | 20% | 3.47 ± 0.09 a | 3.25 ± 0.05 c | 3.78 ± 0.11 c | 2.92 ± 0.13 c | <10 | <10 | + | + |
| 14 | 0% | 4.58 ± 0.23 a′ | 3.14 ± 0.07 a′ | 5.33 ± 0.11 c′ | 3.60 ± 0.11 b′ | <10 | <10 | Nd | nd |
| | 5% | 5.29 ± 0.12 c′ | 3.42 ± 0.20 b′ | 4.01 ± 0.08 a′ | 3.69 ± 0.13 b′ | <10 | <10 | Nd | nd |
| | 10% | 4.60 ± 0.06 a′ | 3.10 ± 0.05 a′ | 4.95 ± 0.05 b′ | 3.30 ± 0.09 a′ | <10 | <10 | Nd | nd |
| | 15% | 5.01 ± 0.08 b′ | 3.47 ± 0.09 b′ | 4.92 ± 0.08 b′ | 3.90 ± 0.04 c′ | <10 | <10 | Nd | nd |
| | 20% | 4.60 ± 0.14 a′ | 3.60 ± 0.11 b′ | 5.47 ± 0.07 c′ | 3.98 ± 0.01 c′ | <10 | <10 | + | + |
| 21 | 0% | 5.65 ± 0.14 C | 3.11 ± 0.13 A | 4.18 ± 0.12 A | 3.91 ± 0.08 C | <10 | <10 | Nd | nd |
| | 5% | 5.87 ± 0.09 C | 3.36 ± 0.07 A | 4.05 ± 0.11 A | 3.86 ± 0.09 C | <10 | <10 | Nd | nd |
| | 10% | 4.27 ± 0.21 A | 3.94 ± 0.11 B | 3.85 ± 0.09 A | 2.89 ± 0.11 A | <10 | <10 | Nd | nd |
| | 15% | 4.81 ± 0.22 B | 3.98 ± 0.12 B | 3.90 ± 0.04 A | 3.42 ± 0.14 B | <10 | <10 | Nd | nd |
| | 20% | 4.90 ± 0.09 B | 4.33 ± 0.08 C | 4.43 ± 0.12 B | 4.32 ± 0.22 D | <10 | <10 | + | + |

Explanatory: TVC—total viable counts, Y&M—total yeast and mold counts, BAC—*Bacillus* spp.; (+)—the presence of *Bacillus* spp. in 10 g of product. Values in the same column marked with the same symbols mean no statistically significant differences ($\alpha$ = 0.05).

Many bakery products can be spoiled by different microorganisms, including fungi and bacteria. The fungal contamination is quite common in raw bakery materials; however, it is not considered as the most critical issue, as the life cells of microorganisms can be destroyed by the baking temperature. On the other hand, the postbaking contamination (air, product handling, equipment sanitized) seems to be an important issue in terms of the microbiological safety of bread [55]. For example, in the Morassi study [55], the raw materials of wheat flour and cornmeal exhibited the fungal counts of approx. $10^2$ log cfu/g in 60% of the tested samples. However, in the other studies, the fungal counts have been reported as higher, at approx. $10^5$ log cfu/g in similar samples [56,57]

Among bacteria contamination, *Bacillus* spp. is claimed as the main agent responsible for a spoilage process known as ropiness or rope. Rope spoilage occurring due to *Bacillus*

bacteria may produce spores and therefore survive during the thermal processing of bakery products. Rope takes place when counts of *Bacillus* spores reach $10^3$ spores/g. Once spores withstand baking, they may further germinate and grow [58]. A good solution for getting rid of *Bacillus* spp. can be the use of sourdough or natural substances like LAB-based bioingredients, which can lower the pH of dough and therefore decrease the thermal resistance of *Bacillus* cells and help in the inactivation of the microorganism through heat treatment [59]. In our study, we do not use sourdough in bread production, as the LAB count was < 10 log cfu/g in all samples during the whole storage period.

From a microbiological point of view, the storage conditions (vacuum or air) used as the association to control the microbiological risk and stability of the bread samples had the biggest influence, and therefore we recommend vacuum as the better solution.

### 3.9. PCA Analysis

A principal component analysis (Figure 9) of the results of the evaluated bread samples showed that the sample variation corresponds to the first main component (Factor 1), which accounted for 54.97% of the total variability and was related mainly to chewiness, elasticity, cohesiveness, crumb features, browning index, foreign and yeast smell as well as gumminess. The second component (Factor 2) constituted 37.87% of the general variable and was related mainly to tastes (yeast, sour, salt and foreign). Based on the obtained eigenvalues, the analysis can be limited to two factors explaining 92.84% of the total variability (Figure 9a).

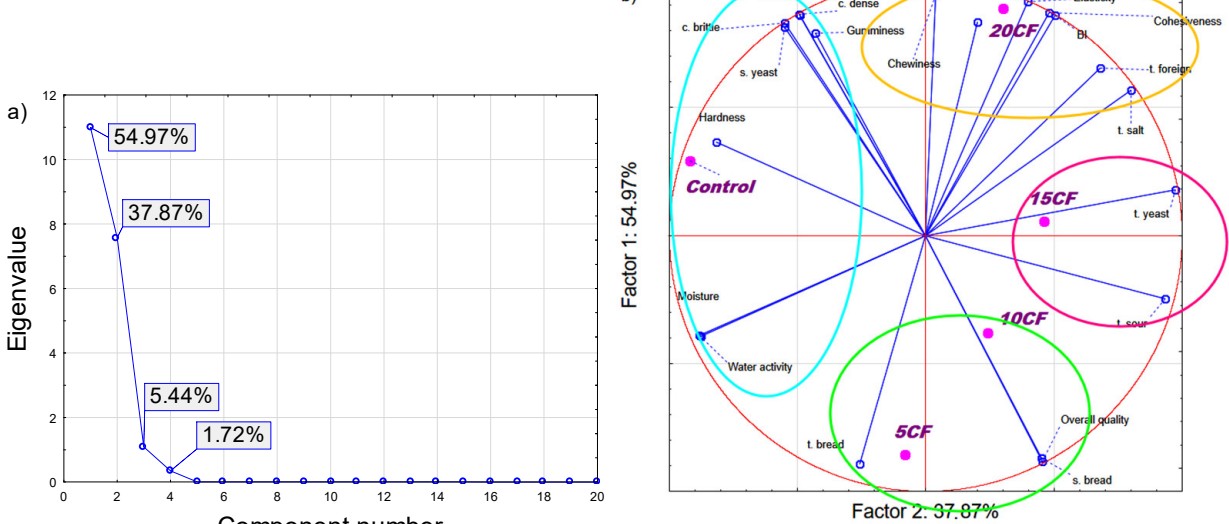

**Figure 9.** Principal component analysis (PCA) of the following samples: Control—control sample, gluten-free breads with the addition of chestnut flour in the amount of 5% (5CF), 10% (10CF), 15% (15CF) and 20% (20CF). BI—browning index, 'c.' stands for crumb, 's.' stands for smell, t. stands for taste. Eigenvalues for each individual principal component is presented in (**a**), but (**b**) presents importance of chosen factors.

The PCA results (Figure 9b) showed that the analyzed samples can be clustered into four distinctive groups. One of them consists of the control sample. The second cluster contains the breads with the 5% and 10% addition of chestnut flour. The next one contains the bread with addition of 15% of chestnut flour, and the last cluster consists of the bread sample with the 20% of chestnut flour addition.

The obtained clusters distinguish the samples with different amounts of chestnut flour additions. The control samples were characterized by the highest level of hardness, moisture and water activity. Additionally, the crumb was brittle, hard and a yeast smell was recognized. One can notice that control samples were accompanied by a higher water content in comparison with samples with the chestnut flour addition.

The addition of chestnut flour in amounts of 5% and 10% showed better organoleptic features than the other samples, such as smell and taste. The bread smell and taste were the most intense in such samples. As a result, both samples were recognized as the most demanding breads among the analyzed ones.

The increase of the addition of chestnut flour up to 15% caused more yeast and sour taste in the analyzed samples. The higher addition of chestnut flour resulted in a faster browning index, and a foreign taste and smell appeared. The bread samples were also characterized by higher level of chewiness, elasticity and cohesiveness.

Based on the provided analysis, one can conclude that the bread samples with different levels of the chestnut flour addition were characterized by different factors. The addition of the chestnut flour in small amounts decreased the level of the browning index, foreign taste and smell, and the level of the moisture. It also increased the feeling of the bread taste and smell in comparison to the control sample. Unfortunately, further increasing the amount of the chestnut flour addition did not improve the level of consumer demands. Samples with chestnut flour additions in the amount of 5 and 10% were better noticed by consumers than the other samples. The only problem is with the elasticity and crumb parameters, because the samples with the best quality are far from the expected values. One can conclude that the obtained results suggested that the addition of chestnut flour in amounts between 5 and 10% have promising organoleptic and sensory features for gluten-free breads.

## 4. Conclusions

Among the tested samples, the best physicochemical parameters were characteristic for breads with the 5% and 10% addition of chestnut flour. The least acceptable parameters, and at the same time the least acceptable quality, were noticed in the bread with the 20% addition of chestnut flour. The addition of chestnut flour in an amount up to 10% to baking gluten-free bread causes an increase in the volume of the tested breads.

The highest porosity of the crumb was characteristic for the bread with the 10% addition of chestnut flour, which also meets the requirements specified in the standards for traditional wheat bread. Moisture analysis showed that the tested gluten-free breads turn stale in a similar way. After 7 days of storage, the humidity of the crumb decreased for all analyzed variants packed with air and vacuum. The water activity for all tested loaves was characterized by the highest values at point zero. The TPA texture analysis showed that the addition of the chestnut flour to the recipe reduced the hardness of the gluten-free bread crumb. As the amount of the added chestnut flour increases, the elasticity and cohesiveness of the bread crumb increases. Bread with the 5% addition of chestnut flour was characterized by the lowest gumminess and chewiness.

As a main finding of the conducted research, we observed that the addition of chestnut flour to the recipe affects significantly ($p < 0.05$) the texture of the finished product, reducing the hardness and increasing the elasticity and cohesiveness of the bread crumb.

It is worth noticing that the use of chestnut flour in an amount of up to 10% increased significantly ($p < 0.05$) the volume of the resulting loaves.

Microbiological research has indicated vacuum packaging as a better way to protect and store gluten-free bread.

The sensory evaluation showed that the bread with the 5% addition of chestnut flour had the best taste and smell. This variant also obtained the highest marks in the consumer desirability survey.

For practical use in future production, it is recommended to replace corn starch in gluten-free breads by no more than 10% by chestnut flour.

**Author Contributions:** Conceptualization, K.M.-L.; methodology, K.M.-L. and D.Z.; validation, K.M.-L., A.Z., M.S., D.Z. and P.L. (Piotr Lukasiak).; formal analysis, P.L. (Patrycja Lesniewska) and D.Z.; resources, K.M.-L., M.S. and K.Z. data curation, K.M.-L., D.Z., P.L. (Piotr Lukasiak) and A.Z.; writing—original draft preparation, K.M.-L.; writing—review and editing, K.M-L., P.L. (Patrycja Lesniewska), M.S., K.Z., D.Z., P.L. (Piotr Lukasiak) and A.Z.; visualization, K.M-L. and D.Z.; supervision, K.M-L.;

statistical analysis, P.L. (Piotr Lukasiak). All authors have read and agreed to the published version of the manuscript.

**Funding:** This research was funded by Warsaw University of Life Sciences.

**Institutional Review Board Statement:** Not applicable.

**Informed Consent Statement:** Not applicable.

**Data Availability Statement:** Not applicable.

**Conflicts of Interest:** The authors declare no conflict of interest.

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
