# Peer review of "The Influence of Chestnut Flour on the Quality of Gluten-Free Bread"

_applsci, doi:10.3390/app12168340_

Round 1

Reviewer 1 Report

The papers describes the effect of different amounts of chestnut flour added to a corn/potatoe starch based gluten-free bread quality. The results are shown clear, however it is a lot of text for this information and results presented in tables are repeated and listed  in the text again, what is not necessary.

line 84: Is chestnut flour in your country an regulated food additive or just an ingredient?

line 110 f.: The amounts of milk and tap water are not mentioned. The amount of liquid added to gluten-free batters/ doughs is a key factor determining resulting bread quality.

Table 1: Please add some information the starches used. There are hundreds of different corn starches on the market with different gelatinization maxima, native or processed, waxy, etc.

l 149-156: Please give also the time between the first and the second compression. This has strong effects on all parameter derived from it,  elasticity

l. 167 instant noodles?

were the bread baked in tins? Which size? or were it hearth breads? What was the dough weight, if bread weight is given in table 2?.

table 4: The subline is not fitting to the content of the table

l. 638 It is not describes, how consumer studies wer performed. Only with celiac patients? Or just the attractivy to the sensory panel (probably not so familiar with quality of gluten free bread )

l. 648f: What was the aim of performing microbiology analysis? It gives just a status on the hygiene / contamination possibilities in the university bakery. Or is chestnut flour rich in bacilli spores? The results in table 5 are log cfu/g ? And it is well known, that excluding air is slowing down the growth of moulds.

Author Response

Dear reviewer,

Thank you for your time and valuable comments you attached to revise our manuscript. In depth analysis and careful examination of our paper without any doubts, increased the level of our work. Below you can find detailed answer for all raised issues. We agree and understand all of them, but in some cases the justification of our point of view has been included. We hope that the revised version of the manuscript will fulfill your comments and be ready for publication. In attachment we addressed your comments point by point.

Reviewer 2 Report

The study on non-gluten bread using chestnut flour, a novel component, is fascinating. The manuscript is written well. However, the manuscript contains a lot of jargon. The jargon in the manuscript needs to be reduced, as do the phrases that use it. We don't use phrases like "one observed that" in academic writing (line 22).

According to Lines 25–26, the microbiological findings suggested vacuum packaging as a more effective method of safeguarding and preserving gluten-free bread... Justify why vacuum package? why not packing for modified atmospheres.

Lines 86–87: Justify why pick chestnut flour out of so many other ingredients? What qualities does it have that help the bread taste better and improve the texture?

In the literature, there are quite a few of past studies using chestnut flour in bread making. The novelty of study need to be included.

Statements in lines 89 and 92 are in conflict.

Chestnut flour is positively influencing the physical and nutritional health, according to line 89.

whereas lines 91–92 stated using too much chestnut flour could compromise the final product's quality . Please specify whatever aspect of quality you are referring to and include information from earlier studies on the effects of using chestnut flour in baked items.

Method (line 173): Describe the storage study's room temperature. Justify your decision to exclusively research storage for 21 days at room temperature.

Interpretation: More discussion on the results is required. The causes of quality variations in the products under investigation during storage are not explained by authors. This has to be completed.

The authors' conclusion based on the significant findings must be included.

Author Response

(The authors gave the same response as above.)

Reviewer 3 Report

General comments: The manuscript is well written and presents some interesting findings. The hypothesis is well described and the language is easy to understand. However, there is a scope for improving its quality and impact. My suggestions in this context are as follows-

Title: As in the present study, the chestnut flour and packaging were also studied, so I would suggest to include packaging in the title.

Abstract:

       i.          There is a need to describe the packaging method?,

     ii.          Line 20: please mention the temperature and storage conditions such as ambient or refrigerated, packaging materials etc

   iii.          Line 21-22: may be deleted from water--------were performed

   iv.          Line 22-25: Please indicate p value at the relevant place

     v.          Conclusion and recommendation: Good

Keywords: please add storage conditions/ packaging

Introduction: The hypothesis is well described and sufficient background information is provided to start the work. However, line 84-85- ---positively influence-----bread; plz cite reference or delete.

Materials and methods

·       Line 111- tap water may be changed to potable water

·       Line 117: based on our own research may be replaced with based on our preliminary findings or trials

·       Room temperature- may be replaced with ambient temperature. Please mention how the temperature mentioned was regulated or the place has same temperature range as mentioned (in later case no need to write the temperature regulation)

·       Sensory analysis: please add more details, sample preparation, coding etc

·       Line 185: p 0,05 should be 0.05, please also mention the sample size

Results and discussion

·       Please mention the composition of chestnut flour (if available for better clarity) such as sugar, protein, dietary fibers etc

·       Plz while describing results and discussion, kindly avoid repetition of data

·       Line 211-correct range may be written as optimum range or desired range

·       Line 239: worst may not be scientific, please replace it with lowest or least preferred or other suitable words.

·       Line 293: please describe the reason of the variation in b color trend, first low then increase, it is the at high content of chestnut flour dominating factor behind b value.

·       Zero point: I assume it refers immediately after preparation?

·       Plz a clarification why the bacterial counts with 5% chestnut flour higher than 10% and Control. Is it due to processing?

The bacterial name should be in italics.

In Table footnote: need to mention what row wise or column wise superscript indicates.

Author Response

(The authors gave the same response as above.)
